# ABWiSE v1.0: Toward an Agent-Based Approach to Simulating Wildfire Spread

Jeffrey Katan[1] and Liliana Perez[1]

[1]Laboratory of Environmental Geosimulation (LEDGE), Department of Geography, Université de Montréal, Montreal, QC, Canada, 1375, Avenue Thérèse Lavoie-Roux, Montréal (QC), H2V 0B3, +1 (514) 343-8003

**Correspondence:** Jeffrey Katan (jeffrey.katan@umontreal.ca)

**Abstract.** Wildfires are a complex phenomenon emerging from interactions between air, heat, and vegetation, and while they are an important component of many ecosystems' dynamics, they pose great danger to those ecosystems, and human life and property. Wildfire simulation models are an important research tool that help further our understanding of fire behaviour and can allow experimentation without recourse to live fires. Current fire simulation models fit into two general categories: empirical models and physical models. We present a new modelling approach that uses agent-based modelling to combine the complexity possible with physical models with the ease of computation of empirical models. Our model represents the fire front as a set of moving agents that respond to, and interact with, vegetation, wind, and terrain. We calibrate the model using two simulated fires and one real fire, and validate the model against another real fire and the interim behaviour of the real calibration fire. Our model successfully replicates these fires, with a Figure of Merit on par with simulations by the Prometheus simulation model. Our model is a stepping-stone in using agent-based modelling for fire behaviour simulation, as we demonstrate the ability of agent-based modelling to replicate fire behaviour through emergence alone.

## 1 Background

Fire is an integral part of ecosystems the world over, but also poses a serious danger to human life and property (Bowman et al., 2011; Moritz et al., 2010; Brenkert-Smith et al., 2013; Butry et al., 2001; Carroll et al., 2006; Chuvieco et al., 2014; Kochi et al., 2010; Richardson et al., 2012). In recent years, anthropogenic climate change has exacerbated this danger chiefly by lengthening growing seasons and increasing the risk of drought (Flannigan et al., 2016; Lozano et al., 2017), leading to more frequent and more extreme fires in many parts of the world (Chuvieco et al., 2016; Kirchmeier-Young et al., 2019, 2017). The use of controlled burning has, for a very long time (Gott, 2005; Roos et al., 2021), helped to mitigate the risks of extreme fires and to maintain forest health (Boer et al., 2009; Camp and Krawchuk, 2017; Paulo M Fernandes and Hermínio S Botelho, 2003). Given the exacerbation of conditions ripe for extreme fires, it is paramount to predict how a fire might spread if it starts, especially for prescribed burns. Fire behaviour models are an important research tool that help further our understanding of fire behaviour and can allow experimentation without recourse to live fires (Hoffman et al., 2018). More specifically, modelling at the scale of individual fires is important for both the study of fire regimes (Keane et al., 2013; Parisien et al., 2019) and the operational management of active fires (Finney, 1999; Lawson et al., 1985; Tymstra et al., 2010; Van Wagner, 1974).

Bearing in mind that a complex system is one in which numerous elements interact in ways that give rise to emergent behaviour, often non-linear in nature, usually featuring feedback loops (Batty and Torrens, 2005; Langlois, 2010), at its base, fire is a complex system of interactions between fuel, oxygen, and heat (Byram, 1959). The dynamics and emerging behaviour are the result of self-organization and the complex system will exhibit some form of hierarchy (Langlois, 2010), e.g. heat released from combustion warms neighboring material to the point of combustion and creates convective currents in the air,

moving oxygen through the system, which in turn keeps feeding the fire (Anderson, 1969; Byram, 1959). In a forest fire, the heat flux of all the burning material contributes to convection in the air mass surrounding a fire, sometimes enough to alter the flow of air that drives it (Clements et al., 2019; Filippi et al., 2009). This fire leaves a portion of land bereft of vegetation until it is recolonized, and this patch of land responds differently to new ignitions thereafter (Parks et al., 2015). Many fires over many years may affect the climate, which affects the vegetation, affecting the fires as a result, forming a feedback loop (Bowman

et al., 2014; Stevens et al., 2015; Stralberg et al., 2018). Nevertheless, the challenge of any fire model is to balance complex behaviour with speed of computation, at a relevant scale.

Our goal is to demonstrate the potential of Agent-Based Modelling (ABM) for the simulation of forest fire spread. Using ABM and a complex systems approach, we build a model that uses simple rules to reproduce fire behaviour as an emergent property of interactions between numerous agents representing fire. Agent-Based Modelling is a useful tool for modelling

complex systems and is broadly much more computationally efficient at reproducing these systems than classical approaches based on solving numerous partial differential equations (Parunak et al., 1998; Sun and Cheng, 2005). As presented in the literature review below, ABM has appeared very little in fire behaviour research, and with this study, we aim to illustrate the potential of this approach to the field of forest fire disturbances and address some of its limitations.

## 1.1  Fire behaviour models

Bearing in mind that wildfires are a global phenomenon that pose significant and growing threats to human lives, property, wildlife habitat, regional economies and global climate change, a variety of tools to tackle and envisage fire propagation have been developed. Some of these tools have the purpose of monitoring (Chu and Guo, 2013; Chuvieco et al., 2019; Giglio et al., 2016, 2003), others to forecast the likelihood of wildfire events (Cheng and Wang, 2008; Taylor et al., 2013; Yue et al., 2018), and lastly some to model and simulate fire behaviour (Sullivan, 2009a, b, c). The literature concerning this latter category is of

particular interest to the goal of this study.

There are many fire behaviour models described in the literature, ranging from empirical relations between environmental factors and fire behaviour to physics-based models that simulate the heat transfer of combustion between fuels and between fuel and atmosphere (Sullivan, 2009a). Among the most important advantages of empirical simulation models is their speed of computation; by simplifying the interactions between environmental factors and the fire front, they only have a small set

of equations to solve at each time step (Sullivan, 2009b). The primary design goal for empirical models is operational use by firefighters, who need rapid results and have the expert knowledge to overcome model limitations (Finney, 2004; Stocks et al., 1989). On the other hand, physical simulation models are better able to represent fire-atmosphere interactions and replicate the complexity and emergent behaviour of real fires (Coen, 2018). For example, while semi-empirical models such as

FARSITE (Finney, 2004) or Prometheus (Tymstra et al., 2010) assume that fire shape is elliptical, physics-based models do not make this assumption and fire shape matches observations through emergence (Filippi et al., 2009; Linn and Harlow, 1998). The drawback of physics-based models is their computation time; since they typically simulate interactions at a very small scale and have huge computational requirements, such models struggle to perform faster-than-real-time (FTRT) simulations (Sullivan, 2009a).

Modellers are attempting to bridge this gap between complexity of model behaviour and execution speed in various ways. One is by coupling computational fluid dynamics models (CFD) with empirical fire behaviour models (Coen et al., 2013; Filippi et al., 2013), though it is argued that the generally coarse scale of the fire behaviour component limits their use (Linn et al., 2020). Yet despite simplifying the fire spread component of a coupled fire-CFD model, FTRT simulation can be difficult to achieve. Using the WRF-Fire (Weather Research and Forecasting – Fire) model (Coen et al., 2013) to simulate a real fire event in Bulgaria, Jordanov et al. (2012) reported simulation speeds based on number of processing cores and noted that FTRT simulation required a minimum of 120 cores. The CFD was the more demanding component of those simulations. It is possible to simulate fire-atmosphere interactions without using a complicated CFD, but instead using a model that considers only relevant airflow. Hilton et al. (2018) create a model of pyrogenic potential to simulate two-dimensional airflow at the fire line and their results match well with real-world experimental fires. While physics-based models provide the most realistic representations of fire behaviour, simplified physical or empirical models are also able to reproduce reasonably realistic fire behaviour by retaining relevant fire-atmosphere interactions. Other models take advantage of principles from complex systems modelling, where complex phenomena are simplified by spreading calculations to individual, interacting computational units known as automata or agents, in order to capture the essential interactions of a system (Sullivan, 2009c).

## 1.2   Complex systems modelling

In complex systems modelling, there are two broad computational approaches to modelling environmental systems, Cellular Automata (CA) and Agent-Based Model(ing) (ABM). CA are a mathematical representation of a complex system wherein a lattice of cells is subject to a set of rules that determine their state and state information is passed between neighboring cells (Gaudreau et al., 2016; Yassemi et al., 2008). Agent-Based Modeling uses autonomous, interacting agents following a rule set, like in CA. The key differences are that in an ABM, agents are mobile and can be heterogeneous; agents can interact with each other and their environment while moving through it, and different agents can follow different rule sets (Perez and Dragicevic, 2012; Pérez and Dragićević, 2011).

There are numerous CA models of forest fire behaviour. Earlier CA fire models had difficulty simulating correct fire shapes, generally due to grid and neighbourhood shape biases (Tymstra et al., 2010). More recently, CA models on par with popular semi-empirical models have been developed, for example, the model by Ghisu et al. (2015) compares well with FARSITE, and that by Yassemi et al. (2008) does so with Prometheus. Due to their simplicity, CA models find use in dynamic fire-vegetation models, which simulate fire-climate-vegetation interactions over long time spans and at coarse spatial scales (Cary et al., 2006; Gaudreau et al., 2016). However, few, if any, CA models we have reviewed account for fire-atmosphere interactions to inform fire behaviour.

Agent-based modelling often simulates systems where mobile individuals are important, such as predator-prey systems (Grimm et al., 2005), flocks of birds or fish (Oloo and Wallentin, 2017), or insect infestations in forests (Pérez and Dragićević, 2011). Agent-based modelling lends itself well to simulating socio-ecological systems, such as forest management (Ager et al., 2018; Pérez and Dragićević, 2010; Spies et al., 2014) where human decision-making must be modelled. While these examples have so far shown the utility of ABM for simulating decision-making entities, ABM does well with physical systems such as particles or smoke. A recent paper (Smith and Dragicevic, 2018), presents a physics-based ABM of forest fire smoke propagation, including two types of agents, one for fires and one for smoke particles. A single fire agent represents a single fire and produces the smoke agents. The fire agents can be either stationary or move according to a very simplified model of fire spread (2 % of surface wind speed), but fire shape and area are not represented. Because the only aspect of fire behaviour present in the model is smoke production, we do not consider this an ABM of fire behaviour.

The one paper we have found that explicitly claims to be an ABM of fire behaviour is the work by Niazi et al. (2010). It uses a virtual overlay multi-agent system (VOMAS) for validation and verification of their fire spread model, where the VOMAS serves as a simulacrum of measurement points in the simulated forest.

Our literature search has uncovered only two fire behaviour models that match our description of ABM, yet the authors refer to them as CA. The first is the Rabbit Rules model of Achtemeier (2003) that bases itself on some of the principles of complex systems theory as described by Wolfram (2002). The first paper describing the Rabbit Rules model (Achtemeier, 2003) does explain that it is not a CA model and that "each element, a rabbit, is an autonomous agent . . . not constrained by the definition of the underlying grid (raster) domain."; nevertheless, the term ABM does not appear. Later papers that use or mention the Rabbit Rules model refer to it as a CA, masking the fact that it uses a completely different modelling approach (Achtemeier et al., 2012; Achtemeier, 2013; Linn et al., 2020). The Rabbit Rules model recasts the physical and mathematical problems of fire behaviour as a set of rules of "rabbit behaviour" due to the analogical resemblance between fire and rabbit behaviours. Rabbits eat, jump, and reproduce just as fire consumes fuel, passes from fuel element to fuel element or spots, and reproduces as it ignites unburned material. In addition to rules for eating (fuel consumption), jumping (spotting), and reproduction (new ignitions), secondary rules modify these to include the effects of terrain, weather, fuel, and fire-atmosphere feedbacks. The Rabbit Rules model produces a ring shape under windless conditions, and a bowed front in high wind, without any predetermined geography such as in ellipse-based models. Just like an ABM, Rabbits move across the landscape, interact with each other and their environment, and produce reasonable perimeter shapes due to emergence alone.

Achtemeier (2013) presents a field validation of the Rabbit Rules model with the FireFlux experimental grassland fire, conducted in tall-grass prairies near the Gulf Coast of Texas, USA (Clements et al., 2007). The field validation demonstrates a reasonable match between simulated and observed airflow 2 m above the surface at two observation towers used in the FireFlux experiment. That study also notes that the Rabbit Rules model can simulate non-linear processes unachievable by empirical models, and much more quickly than full-physics models; where FIRETEC can take about 90 s for each second of simulation on a 64-processor supercomputer, Rabbit Rules took only 0.67 s for each second of simulation on a desktop PC for this experiment. The simulation speed information for FIRETEC comes from a review by (Sullivan, 2009a). We have not found more recent information on simulation speed for FIRETEC, although the website for HIGRAD/FIRETEC states that

"FIRETEC takes the huge computational resources at the Los Alamos National Laboratory to run, so it is currently a research tool only." (https://www.frames.gov/firetec/home, accessed April 20, 2021)

The initial exploration of ABM applied to fire behaviour by Achtemeier (2003, 2013) provided the base for a new model, QUIC-Fire (Linn et al., 2020). QUIC-Fire is a fuel-fire-atmosphere simulation model that combines the rapid wind solver QUIC-Urb (Singh et al., 2008) with their new physics-based fire spread model Fire-CA. This fire spread model builds on the conceptual framework of the Rabbit Rules model, where instead of "rabbits", energy packets (EPs) represent units of energy that can evaporate moisture, burn fuel, or transfer their energy to the atmosphere. While Fire-CA is described as a cellular

automata model, the EPs act like agents that move across the grid-based computational landscape, therefore we include it with the Rabbit Rules model as the only other example of fire behaviour simulation using ABM. Linn et al. (2020) demonstrate the model in two case studies, comparing the simulation results of FIRETEC and QUIC-Fire. The first case study was a simulated grass fire and the second was a simulated prescribed fire in a forest landscape, replicating conditions typical of a prescribed burn at Eglin Air Force Base, in Florida, USA. Even though the paper does not report simulation speed, it does state QUIC-Fire

is capable of FTRT simulation and required 1/2000 the computational cost of FIRETEC for the simulations reported.

    As stated earlier, the aim of this study is to demonstrate the potential of ABM for the simulation of forest fire spread. To do so we build an agent-based simulation model of fire behaviour using an empirical approach. This allows us to demonstrate how interactions between agents can produce common patterns found in fires by following simple rules. The model proposed here does not aim to replace or upgrade any fire spread model, but rather to showcase the advantages and potential of using an

145 alternative modelling approach. With this in mind, we design it for the simulation of large individual fire events in Canada, as large fires (> 200 ha) account for 3 % of fires in Canada, and are responsible for 97 % of total area burned (Stocks et al., 2002). Fires of this scale are particularly relevant for the study of fire-regimes and fire-climate-vegetation interactions as present in Dynamic Global Vegetation Models. These types of models typically use very simple fire spread models (Keane et al., 2004), and could benefit from a computationally efficient fire spread model that accounts for complex interactions during a fire event.

An ABM of fire spread could potentially fill this niche.

    The proposed and implemented Agent-Based Wildfire Simulation Environment (ABWiSE) model represents the fire front as a set of moving agents whose behaviour is determined by rules accounting for vegetation, terrain, and wind, and the interactions among the agents and with their environment, (such as fire-wind feedback). We implement the ABWiSE model on two base case scenarios and two parts of one real fire (cases 1 through 4, respectively). The first two cases are simulated fires and the

155 latter two are from a fire in Alberta, Canada. The cases are detailed in Sect. 3.1 and listed in Table 1. We calibrate the model with cases 1, 2, and 3, and validate the model against case 4, as well as progression data for case 3. While we perform some preliminary uncertainty and sensitivity analyses to calibrate the model and evaluate some assumptions, a thorough uncertainty and sensitivity analysis will be the subject of future work.

## 2 ABWiSE fire spread simulator

### 2.1 General overview

ABWiSE translates the concept of a moving fire front as a set of mobile fire agents that, viewed in aggregate, form a line of varying thickness. Ultimately, the goal of such a fire simulation model is to provide predictions of the behaviour of hypothetical fires. Presently, this paper uses ABWiSE to explore how ABM, using simple interactions between agents and a simple atmospheric feedback model, can simulate emerging fire spread patterns. Specifically, we aim to identify the strengths and weaknesses of ABM applied to this purpose, and how it differs from other modelling approaches.

We use pattern-oriented modelling as a strategy to both design and evaluate our model (Grimm et al., 2005). The patterns in question are fire line rate of spread (temporal), fire shape (spatial), and fire-wind interactions (emergence). As mentioned earlier, the ellipse is widely accepted as the generic fire shape (Anderson et al., 1982; Van Wagner, 1969), and it serves as the starting pattern. In uniform fuel and wind conditions, how can we get agents to burn an elliptical area through emergence, and not an explicit rule? The guiding assumptions that lead to the model's current form are that fire is slower at the edges of a fire line than the center (Finney, 2004; Van Wagner, 1969), that the relationship between wind speed and rate of spread changes with the angle between the direction of spread and wind direction, and that fire dries the fuel ahead of it, making it more flammable (Byram, 1959). Fire line rate of spread and fire-wind interactions are what create fire shape as it evolves over time (Clark et al., 1996), so we use fire shape to evaluate our model under different conditions and at different times. The specific evaluation scenarios are described in Sect. 3.

### 2.2 Entities, state variables, and scales

The model entities are fire agents and grid cells. The fire agents have two main properties, heading and rate of spread (RoS), plus their location (floating-point coordinates). The heading is the direction, in degrees, an agent faces. The RoS is the portion of a cell an agent travels every time step (called a tick). The units of the RoS depend on the spatial and temporal scales of the model.

Grid cells represent the world with which fire agents interact. Their state variables are their location (integer coordinates, at the center of the cell), fuel level, flammability, slope, aspect, wind speed, and wind direction. Location, slope, and aspect are static throughout a model run, while the other variables can change. Flammability and fuel are unitless variables ranging from 0 to 1. Slope is input as a percentage, where flat is 0 % and vertical is 100 %. Aspect is in degrees. Wind speed is input as km h-1. Flammability is a measure of how quickly the vegetation in a cell can burn, and thus how fast the fire can move through it. Fuel is a measure of the amount of material available for combustion in a cell. Wind can be spatially and temporally uniform or dynamic.

The spatial and temporal resolutions of the model are linearly proportional, e.g. at a 200 m cell size, each time step (or tick) represents 1 minute, and at a 400 m resolution, each tick is 2 minutes. The spatial and temporal extents of the model depend entirely on the scenario to simulate. The model is implemented in the NetLogo multi-agent programmable

modeling environment (Wilenski, 1999). The code and data are freely available under an open source license on GitHub (https://doi.org/10.5281/zenodo.4976112, Katan, 2021).

We base the flammability and fuel of cells on the Canadian Forest Fire Danger Rating System (CFFDRS) (Van Wagner, 1974; Wotton, 2009). We chose it due to the availability of fuel type data in Canada and the system's use around the world (Opperman et al., 2006). The system includes sixteen classes of vegetation for which there are empirically derived equations relating fuel moisture and weather to fire behaviour. The CFFDRS is currently composed of two sub-systems: the Fire Weather Index (FWI), which provides a general rating of fire spread potential based on fuel moisture, temperature, and wind speed, and the Fire Behaviour Prediction system (FBP), which combines the FWI with fuel type characteristics to provide more detailed fire behaviour information. This system has been in use for decades, but does not account for any form of feedback mechanism. Because ABWiSE uses feedback loops to replicate fire behaviour, using the FWI would require reworking its equations without wind, as it is an important variable within its subsystems. Doing so is beyond the scope of this research at this time. Instead, we chose to map the average characteristics of fuel types of the CFFDRS, as described in (Forestry Canada Fire Danger Group, 1992) to flammability and fuel values (Sect. A1). This keeps wind as a separate input and variable that forms part of a feedback loop.

## 2.3 Procedures

Figure 1 provides a schematic overview of the procedures. A model run begins with an ignition, creating four fire agents at that point, each facing one cardinal direction. Since flammability is the first driver of fire spread, fire agents have an initial RoS value set to the flammability of the cell they start in. At each time step, fire-wind interactions provide a local effective wind speed and direction for cells within a certain distance of fire agents (Feedback procedure, Sect. A2.1). Next, fire agents update their RoS and heading based on wind, flammability, terrain, and the local density of fire agents, and then move by that RoS in that direction (Spread procedure, Sect. A2.2). After moving, agents preheat the cell within the distance of their RoS by a small amount, raising its flammability (Preheating procedure, Sect. A2.3). Next, agents have a chance to be extinguished (or die) based on the fuel value at their location and their RoS (Death procedure, Sect. A2.4). Those that do not die then propagate if they have travelled more than a certain distance from their point of origin and if there are fewer than a set number of other fires already in their current cell (Sect. A2.5). Lastly, fire agents reduce the amount of fuel in a cell based on their RoS (Sect. A2.6). Simulation ends if there are no more fire agents, or after a predetermined number of iterations. Detailed descriptions of the procedures, including the equations involved, are included in the Appendix (Sect. A2). However, we will explain here some of the reasoning behind key procedures, namely Feedback and Spread. The Feedback procedure combines the input wind values at a cell with the effect of fire agents nearby and smooths the resulting vector based on the wind in nearby cells. This is a simple proxy for fire-wind interactions that was inspired by the pyrogenic potential of (Hilton et al., 2018). The Spread procedure attempts to match the relationship between RoS and wind speed and direction to observations, as well as producing a reasonable fire shape. In short, low wind speeds have a small effect on the fire agents, but have a stronger effect on fire agents whose heading is close to the wind direction. The relationship between RoS and wind speed follows a logistic curve based on the same assumption as (Forestry Canada Fire Danger Group, 1992) that there exists a maximum RoS based on fuel type,

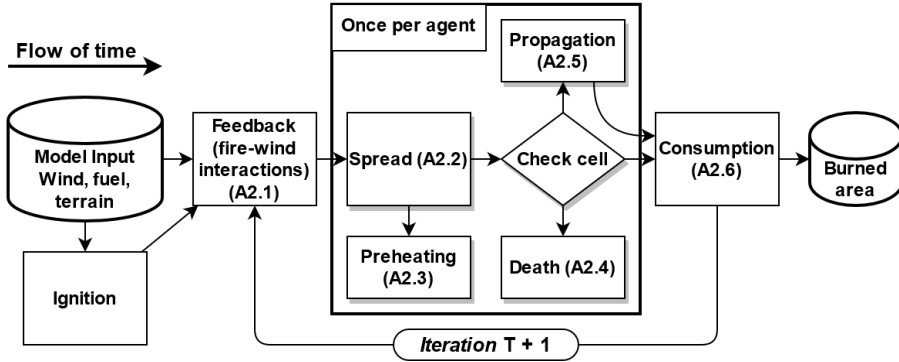

**Figure 1.** Schematic description of model procedures. The "Check cell" diamond represents the check for the **Death** procedure, followed by the check for **Propagation**, after which **Consumption** occurs. Fire agent RoS and heading are updated in the **Spread** procedure, which are used by the **Feedback** procedure at the start of the next iteration. Because agents update asynchronously within a procedure, **Preheating** by one agent can affect the next agent to perform the Spread procedure.

though the relationship is not identical. The various feedbacks change the final RoS from that particular logistic equation, so it would be moot to use the exact same relationship between wind speed and RoS as the FBP system.

The last important detail about model processes is stochasticity. There are two sources of stochasticity in the model: the first is the chance of agents dying out, and the second is the turn order of the agents at each time step. While the operations listed above happen in the order presented, the order of agents or cells performing them is random. This asynchronous updating of agents is a default of the NetLogo programming language and serves to avoid artifacts of execution order. Due to the stochasticity, we base the model evaluation on ensemble maps representing the sum of 100 simulations with the exact same inputs and parameters.

## 3    Calibration and validation

To calibrate ABWiSE, we compare its output with expected behaviour, and adjust the parameters until it performs adequately. Overfitting is a serious problem with this approach and we try to minimize it by fitting our model to three different scenarios. However, detailed fire behaviour data and corresponding weather data are difficult to come by, especially for large and remote wildfires. Fortunately, the free-to-use Prometheus model (Tymstra et al., 2010) offers a sample dataset of a real fire for download: the Dogrib fire of 2001 in the foothills of the Rocky Mountains in Alberta, Canada (Mcloughlin, 2019). We use one part of this fire to for calibration, leaving another part for validation. Because we did not find other datasets using the Canadian FBP fuel type as model input, the two other scenarios for calibration are base cases as simulated by Prometheus.

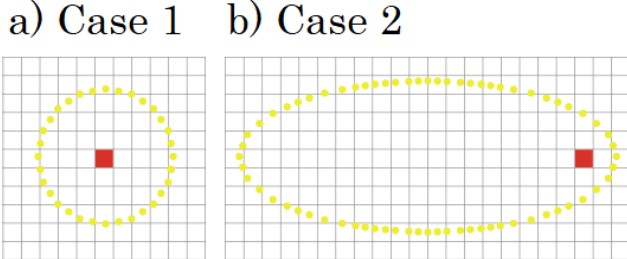

**Figure 2.** The two base scenarios.

**Table 1.** Scenarios used for model calibration and evaluation

| Scenarios | Description |
|-----------|-------------|
| Case 1 | C-2 (Boreal Spruce) fuel type, no wind |
| Case 2 | C-2 (Boreal Spruce) fuel type, 20km E |
| Case 3 | Dogrib Fire, Oct 16 portion |
| Case 4 | Dogrib Fire, Sep 29 portion |

### 3.1 Scenarios

The four scenarios are listed in Table 1 for ease of reference. The first base case scenario (Fig. 2(a)) is a 2 hour-long fire on a flat plane of the C-2 fuel type, Boreal Spruce, with no wind, and temperature of 25° C. The second base case (Fig. 2(b)) is the same but with 20 km/h wind speed, coming from the East. Though ABWiSE does not use temperature as an input, Prometheus uses it to calculate the FWI and track changes in fuel moisture over time.

The Dogrib fire started on 25 September 2001 in the Rocky Mountains of southwest Alberta, Canada. The fire was detected at 17:00 MDT on September 29, and reached a size of 675 ha at 16:30 the next day. Fire suppression started at 6:00 MDT on October 1. It burned at various rates under some suppression efforts until it grew to 852 ha by October 15. On the 16th, a wind event pushed the fire through a gap in the surrounding mountains and caused the fire to jump the Red Deer River. After this, it spread 19 km in 6.7 hours in a northeast direction. The final fire size was 10 216 ha, 90 % of which was a result of the October 16 fire run (Mcloughlin, 2019). The vegetation consumed by the fire consisted mostly of lodgepole pine (*Pinus contorta*), followed by subalpine fir (*Abies lasiocarpa*) and Engelmann spruce (*Picea engelmannii*). Respectively, the FBP fuel types C-3, C-1, and C-2 represent these.

The example scenario provided with Prometheus provides data for both the initial unsuppressed burn between 17:00 on September 29 and 18:30 on September 30, and the October 16 fire run, shown in Fig. 3. These represent cases 4 and 3, respectively. Case 4 serves as an independent dataset for validation, and the final perimeter of case 3 serves for calibration, while the progression perimeters (solid polygons in Fig. 3, as provided with example data) are used for interim validation. The nearby Yaha Tinda automated weather station provided the necessary weather data for the simulation. According to the

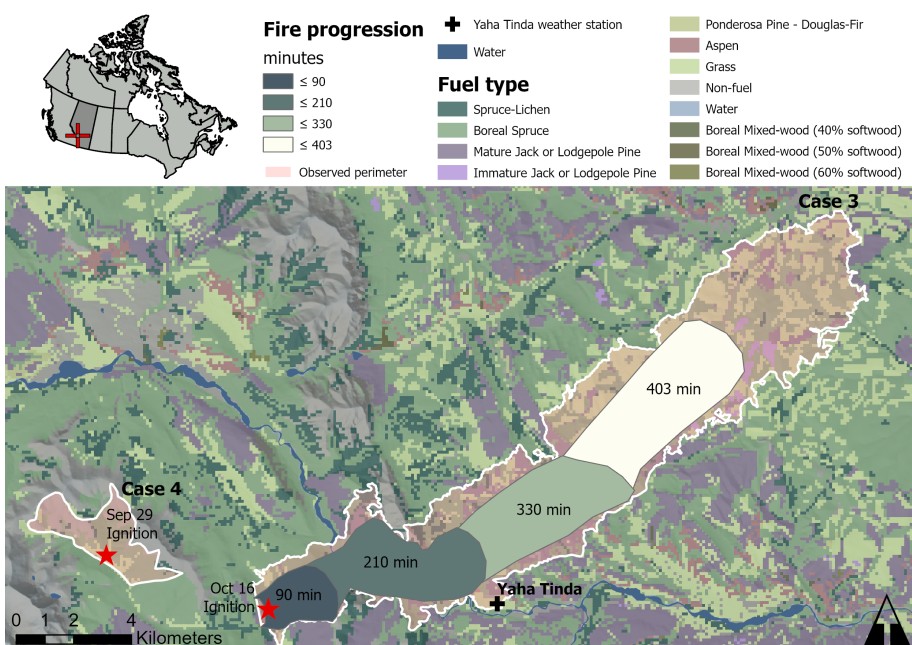

**Figure 3.** Two parts of the Dogrib fire. The ignition points are those provided with the sample data, as used in our simulations as well as the Prometheus simulations. Background is a combination of hill shading, elevation, and fuel-type. The solid polygons show fire progression representing the time by which at least that much area has burned.

case study, this single weather station could not account for the complex topography of the mountainous area. The Dogrib case study includes a manually created weather patch for the Prometheus simulation in order to replicate the wind funnelling effect of the Red Deer River valley observed in the actual fire event. This funnelling is what drove the fire through a gap in the mountains and across the river. The report states that use of this weather patch provided more realistic simulation results in the case study than either uniform winds or dynamically modelled weather grids accounting for topographical influence on wind flow (Mcloughlin, 2019). We use the exact same weather information for the ABWiSE simulations.

After calibration and validation, we perform a preliminary sensitivity analysis of the model's response to different fuel types by repeating the experiments for the cases 1 and 2, but with the other fuel types present in the fuel type map for the Dogrib case study, as well as testing case 3 on a randomized fuel map.

## 3.2   Model evaluation

Evaluation is critical for any simulation model, especially one that relies on empirical relations between variables instead of physical rules. Quantitative spatial methods to measure fire behaviour model performance broadly fall into two categories: final perimeter methods and time-based methods. Final perimeter methods, as the name suggests, measure the similarity between the final simulation perimeter and a final observed perimeter. Such methods are dependent on the error of the observation time,

related assumptions about simulation duration, and they provide no information about model performance at intermediate times (Filippi et al., 2014).

In order to measure model performance throughout calibration, we use a final perimeter measure: the Figure of Merit (FoM) (Eq. (1)), equivalent to the Jaccard similarity coefficient (Pontius et al., 2018). Values of the FoM range from 0 to 1, with 1 being a perfect match. In the case of fire perimeters, hits are those cells burned by the simulation that were also burned in the observation, misses those unburned by the simulation, but burned in observation, and false alarms those burned by the simulation that were not burned in observation. Though there are some criticisms about the FoM and its use in measuring

land-use change models (Harati et al.; Varga et al., 2019) it still provides useful information and is easy to interpret when used to compare burned areas (Filippi et al., 2014). In particular, criticisms surrounding the FoM are based on full map comparisons, but in this study, the comparison is between burned perimeters only. Correct rejections far from the area of interest are never considered. Furthermore, simplicity of calculation is an important factor when measuring millions of simulations, as necessary in calibrating this model.

$$FoM = \frac{hits}{hits + misses + falsealarms} \qquad (1)$$

## 3.3   Calibration

Calibration begins with manual exploration of the parameter space, to eliminate parameterizations that produce very inaccurate results (based on both visual assessment and FoM). Large deviations from these manually identified initial settings produce very poor results (e.g. FoM less than 0.2). The next stage explores promising regions of parameter space at finer resolution,

which consists of varying parameters around those manually identified initial settings by steps of about 5 % of their total range, up to 3 steps above and below the initial setting. We uses up to 3 steps because varying all 12 parameters by 6 steps would require just over 2 billion simulations. In addition, the need to repeat simulations for each combination to account for stochasticity acts as a multiplier to that number. We keep our parameterization runs to about 100 000 combinations at a time, with three repetitions, choosing to vary most those parameters deemed to have the most impact, and with those fixed varied

the other parameters. The parameterization runs generate a table with each row containing the parameter values and the final FoM of that simulation. After such a run, we use a classification and regression tree (CART) (Brieman et al., 1984; Loh, 2011) based on the table to identify new "search" areas for parameterization. That is, the CART identifies important parameters and determines the values above or below which the FoM was better. However, the non-random nature of how we set the parameters and explore parameter space means the CART models are never as robust as if we had used random samples of the parameter

space. We repeat the parametrization – CART process twice to arrive at a parametrization that adequately simulates all three scenarios, without over-fitting the model to the few scenarios available to us.

    After calibration, we use Monte Carlo methods (Kroese et al., 2014; Metropolis and Ulam, 1949) to account for the stochasticity of the model, producing ensemble maps of 100 simulations of each scenario. One ensemble map is the sum of the output maps of all 100 simulations. Cell values in these maps range between 0 and 100, representing how many times it burned in the

305 ensemble, or, in other words, burn probability. Note that this is the probability of that cell burning in the ensemble of simula-

tions, not a prediction of burn probability in reality. We calculate a more robust FoM based on the statistics of these probability maps.

## 3.4 Validation

Typical model validation compares model output for scenarios not used in model calibration (Hoffman et al., 2018). The only independent scenario available is case 4, the September 29-30 portion of the Dogrib fire that was not subject to suppression efforts. This fire was about 14 km away from the automated weather station that supplied the data for the Dogrib case study, and nestled in a mountain valley. The available weather data are almost certainly less accurate than those for the Oct. 16 run, which had improvements from experts and field observations. Therefore, while independent for the sake of model validation, the quality of the data limits the robustness of this validation. We supplement this validation with a time-based method to validate interim fire behaviour. Because the measure for calibration was only final perimeter, the measure of intermediate fire behaviour is to some extent independent and can validate interim behaviour (Filippi et al., 2014). Although this cannot validate the whole model, it helps to elucidate the validity of its mechanisms. In addition to burn probability maps, the ensemble simulations provide maps of mean arrival time, which we use with the Dogrib fire progression data for case 3 in a time-based measure of performance to validate interim behaviour. Progression data for case 3 consist of reconstructions of the Dogrib fire perimeter at four instances between the start and end of the fire. Progression data for case 4 are too sparse for this method. Details for this evaluation are in Sect. 4.3.

## 4 Results

### 4.1 Ensemble maps

The ensemble maps in Fig. 4 provide a visual overview of model performance. Cases 1 (Fig. 4(a) and (b)) and 2 (Fig. 4(c) and (d)) both show an excellent agreement between simulated and expected shapes, with case 2 having slightly more variation in its final perimeter. Since there is wind driving the fire in case 2, there is more room for emergence through the fire-wind interactions, and thus we would expect more variability in the ensemble simulation. This variability is even more present in case 3 (Fig. 4(e)), where not only is the time of burn longer, but the input wind itself is more dynamic. The ensemble simulations of case 3 demonstrate a wide range of potential outcomes. The majority of simulations do burn within a similar area as the real Dogrib fire, though some also cross the mountains further north and burn a large swath of land parallel to the real fire. Simulation of case 4 often burns far less than the observed fire. The simulation also never burns for the full duration of the observation data, with a mean burn time of 421 minutes out of 1550, and a maximum of 1492 minutes before burning out completely. Table 2 summarizes the FoM of the simulations in an ensemble, i.e. the summary statistics of the score of each individual simulation. To test whether the fire-wind feedback has a meaningful influence on fire behaviour in ABWiSE, we also perform these simulations with the $w1$ parameter (see Sect. A2.1) set to 1, which effectively reduces the effect of fire-wind feedback to 0. The table also includes the FoM for simulations by Prometheus, for comparison.

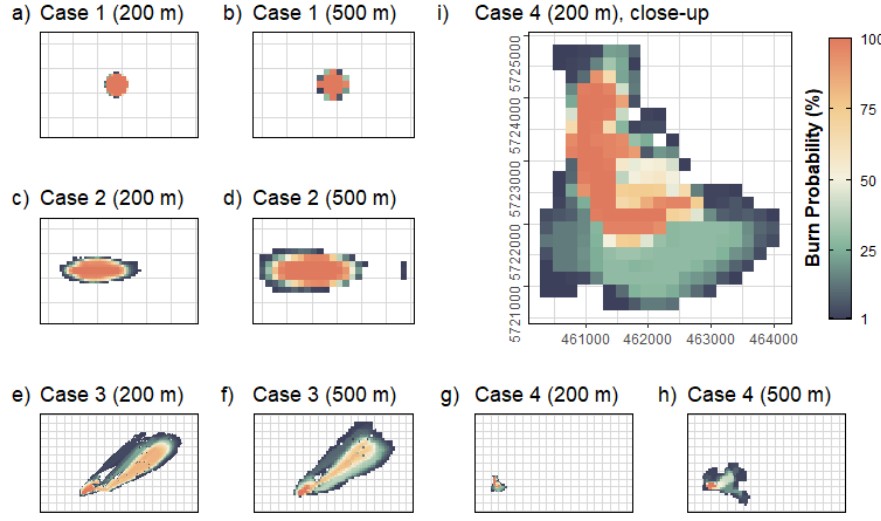

**Figure 4.** Ensemble maps for all 4 scenarios at 2 resolutions (200 m and 500 m). Graticule spacing for every map is 2000 m except (i), for which it is 500 m. This is to help compare the size of each scenario. Sub-figures (e) to (h) have the same spatial extent in order to show the relative size and locations of cases 3 and 4.

Considering that ABWiSE is designed with coarse scales in mind, we perform the same simulation measurements as before, but at a resolution of 500 m per cell instead of 200 m. The scores, in Table 2, are generally lower, due in part to the smaller total number of cells involved in the calculations. As for fire-wind feedback, almost every scenario scores lower without the
340 feedback effect than with it. Only case 1 at a 200 m resolution and case 3 at a 500 m resolution score better without it. Most notably, the maximum scores are all lower. As seen in Fig. 5, simulated fire shape under windy conditions tend to be fan-shaped, rather than elliptical, with Case 2 showing the most evident difference between simulations with and without fire-wind feedback.

### 4.2 Figure of Merit maps

There are two ways of calculating and showing the FoM of these ensemble maps. The first, in Fig. 6, shows an ensemble of the FoM components as calculated for each individual simulation. This visualisation of the different FoM components provides insight into just how, and where, the model agrees or disagrees with observations. For case 1 (Fig. 6(a), (b), (c)), the simulation burns the entirety of the observed burn almost all the time, misses a few cells in a scant 2 % of simulations, and over-burns only a small ring outside the observation perimeter. With a mean FoM of 0.83, the simulation of case 1 is in good agreement
with observations, and notably, creates a circular perimeter through emergence alone. Case 2 has a mean FoM of 0.81, also indicating good agreement. Figure 6(e) and (f) show that there is a fair amount of under- and over-burning in the ensemble, which contribute to the error. Once again, the simulation perimeter closely resembles the expected ellipse through emergence alone. Case 3 has a mean FoM of 0.48, much lower than the first two cases. Figure 6(g) shows that the majority of simulations

**Table 2.** Figure of Merit (FoM) descriptive statistics based on each run in an ensemble simulation. "(no fb)" specifies scenarios run with the fire-wind feedback disabled.

| Case | 200 m resolution | | | 500 m resolution | | | Prometheus |
|---|---|---|---|---|---|---|---|
| | Mean FoM | Max FoM | Stand. Dev. | Mean FoM | Max FoM | Stand. Dev. | FoM |
| 1 | 0.8322 | 0.8800 | 0.0185 | 0.5416 | 0.6300 | 0.0286 | 1 |
| 2 | 0.8080 | 0.9300 | 0.0873 | 0.5598 | 0.6900 | 0.0524 | 1 |
| 3 | 0.4783 | 0.6800 | 0.1830 | 0.4279 | 0.7200 | 0.1839 | 0.5647 |
| 4 | 0.2817 | 0.4100 | 0.0608 | 0.2139 | 0.4600 | 0.0803 | 0.2108 |
| 1 (no fb) | 0.8362 | 0.8769 | 0.0141 | 0.4939 | 0.5882 | 0.0204 | NA |
| 2 (no fb) | 0.4558 | 0.5507 | 0.0431 | 0.3716 | 0.5000 | 0.0274 | NA |
| 3 (no fb) | 0.4116 | 0.4550 | 0.0214 | 0.4449 | 0.5507 | 0.0731 | NA |
| 4 (no fb) | 0.1377 | 0.1806 | 0.0277 | 0.1423 | 0.3673 | 0.0647 | NA |

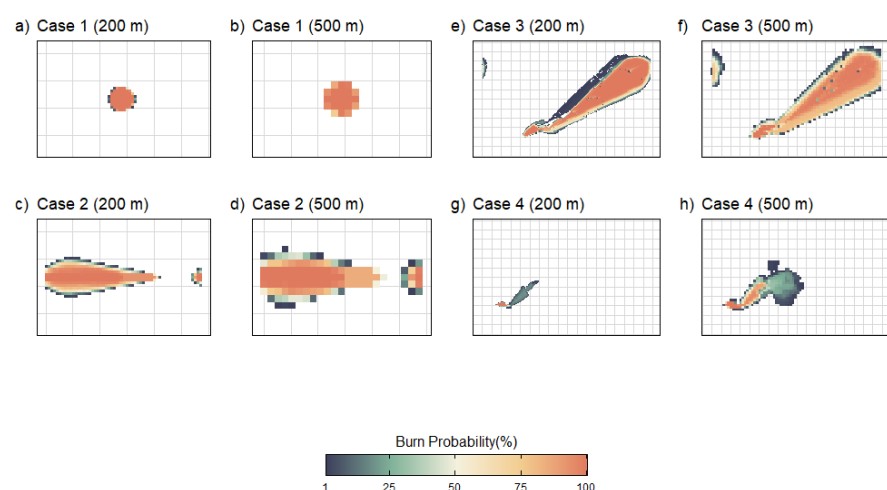

**Figure 5.** Ensemble maps for all 4 scenarios at 2 resolutions (200 m and 500 m) for simulations with no fire-wind feedback. Note that the simulation space topology is toroidal, meaning that fire agents that reach the edge of the world disappear and reappear at the opposite edge. Under normal circumstances this condition should not be reached, but the lack of fire-wind feedback resulted in exceptionally fast-moving fire fronts that passed the edge of the world before the end of the simulation. The results of this can be seen in sub-figures (c), (d), (e), and (f).

in the ensemble do burn a similar shape and area as the observation, but the ensemble frequently under-burns the top edge of the Dogrib fire. Figure 6(h) shows the corollary, and demonstrates that the model very rarely burns the full width of the Dogrib fire, particularly in the bottom portion of the fire. Over-burning is the smaller source of error for case 3, with mostly

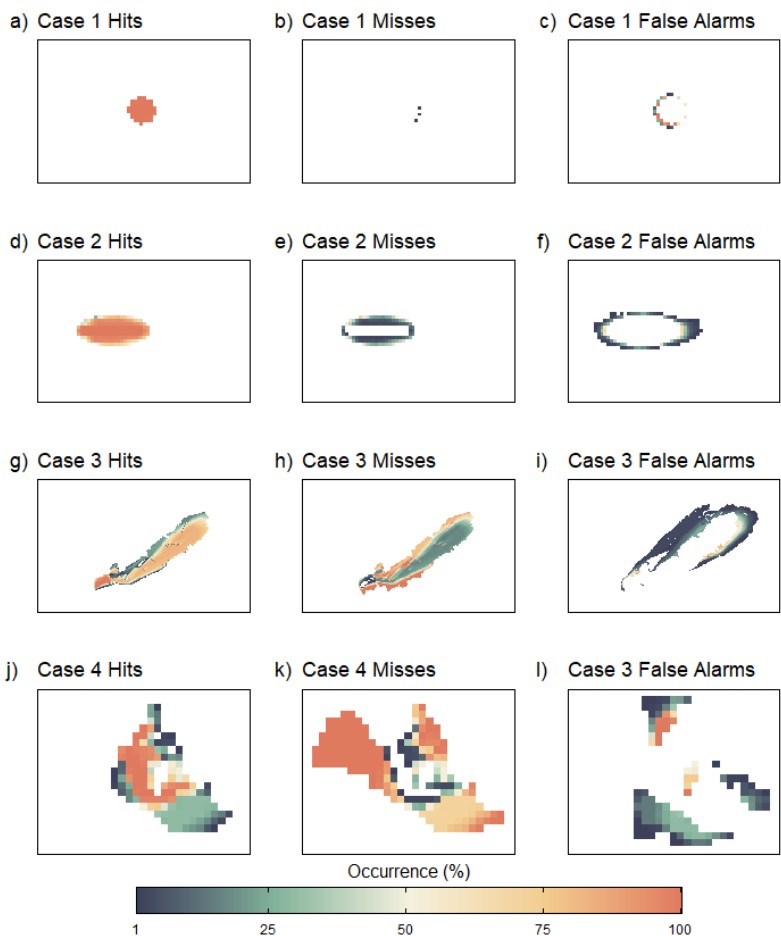

**Figure 6.** Individual FoM components from each simulation, stacked as ensemble maps. Cell value represents how many times a cell was part of a given component in the ensemble, in other words, the percent occurrence in that category. In this case count and percent are equivalent, given an ensemble consists of 100 simulations.

low probabilities, and Fig. 6(j) highlights the particularly low probability of the northern parallel burn mentioned earlier. Case 4 has a mean FoM of 0.28, and never burns the full extent of the observed fire, as visible in Fig. 6(k) and (l). On the other hand, the simulations very rarely over-burn, except a small portion to the North of the fire, which is the top of a ridge.

The second way to calculate FoM from ensemble simulations uses a statistically derived subset of the full ensemble map to calculate FoM. The subset consists of those cells with a burn probability above a certain threshold value, and this subset is compared to the observation data. We calculated the FoM for four threshold values: the mean, one standard deviation, two standard deviations, and the 2nd quartile. This provides the FoM of the most probable outcome of the model. Figure 7 shows the resultant FoM maps and scores for cases 3 (Fig. 7(a) to (d)) and 4 (Fig. 7(e) to (f)) for the four different thresholds. These

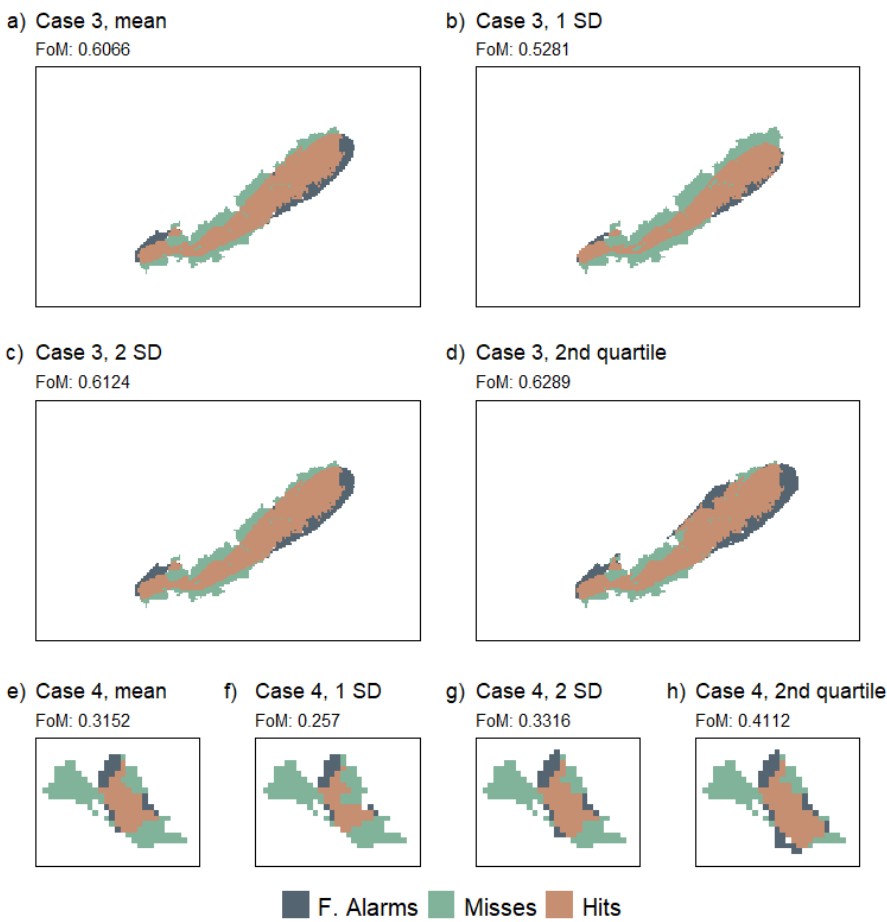

**Figure 7.** Components of Figure of Merit. Each column shows the FoM components and score for cells in the ensemble with a burn probability value above a threshold burn probability. First column is cells in the ensemble with a value above the mean, second column is those above 1 standard deviation below the maximum (100), third is above 2 standard deviations from the max, and fourth is above the 2nd quartile. Note that the real fire shape is the combination of Hits and Misses, while the simulated fire shape is the combination of Hits and False Alarms.

results show that while the mean FoM for cases 3 and 4 are relatively low, the most probable outcomes of the model (as defined by the statistical subsets) score higher.

## 4.3 Time-based evaluation

A time-based measure of model performance allows us to evaluate interim fire behaviour, and thus validate to some extent the processes that make up the simulation. In particular, it reduces the impact on FoM of impossibly burned cells, as mentioned above. Figure 8 shows how our simulation corresponds to the coarse reconstructions of the Dogrib fire (which are included

with the sample data). Simulation progression uses the mean arrival time of a cell to determine those burned within a time period, and we use the mean probability subset (shown in Fig. 7(a) for comparison. In the first time period (Fig. 8(a)), both reconstructed and simulated fires grow similarly, though offset, but their fronts advance to a similar point. In the second period (Fig. 8(b)), the simulation fire continues to over-burn to the north, and lags behind the furthest eastward extent of the reconstructed perimeter. By the third period (Fig. 8(c)) the simulation fire rushes ahead of the reconstruction, though the width of the fires stays similar. At the end of the fire the simulation has burned further and wider than the reconstruction. These tests show a degree of agreement between the progression of the simulated fire and that described in the case study (Mcloughlin, 2019), in that the fire grows slowly in the first two periods, then spreads very quickly in the last two.

## 4.4 Fuel type sensitivity

ABWiSE uses a very simplified representation of fuel types, and it is explicitly calibrated to a single fuel type (C1, Boreal Spruce, with cases 1 and 2), and only implicitly for other fuel types (case 3). Therefore, the assumptions related to the fuel variables remain untested, both for the parameterization of fuel type values and their influence on agent RoS. By repeating the ensemble runs for cases 1 and 2, but with other fuel types, we can determine how well the other fuel types are modelled, (however this does not allow us to distinguish between problems with parameterization of fuel types and parameterization of model equations). The experiments for the sensitivity analysis of fuel type use the same ignition and weather as cases 1 and 2, and change only the fuel type. We test the 9 different fuel types present in the Dogrib fuel type map, and perform the usual 100 simulations to account for stochasticity. To test whether fuel type plays an important role in case 3, we also run 100 simulations of that scenario but with a randomized fuel map. This randomized fuel map only changes the fuel type of cells that already had a fuel type other than non-fuel or water.

As seen in Table 3, model performance varies greatly across fuel types. The model scores a FoM above 0.5 in 11 out of the 18 uniform fuel scenarios, but by fuel type, only 4 out of 9 have such scores at both wind speeds. The model regularly under-burns (i.e. misses) for fuel types C1 and C6, while over-burning (i.e. false alarms) is the larger kind of error in those fuel types for which the model scored poorly. As for case 3 with randomized fuel, ABWiSE almost exclusively under-burns.

## 5 Discussion

Overall, the results of our simulations show good agreement between ABWiSE simulations and observations. The model performs very well in simulating the two base cases, while its performance decreases when simulating the real fire of cases 3 and 4. Ensemble simulation produces an improved score, but also introduces certain problems related to the process-based nature of the phenomenon for case 3 (Fig. 7(b)), where using the 2nd quartile as a threshold includes cells in the subset that could only have been burned if the fire had burned further north earlier in the simulation. The core concept of using ABM to simulate fire spread has proven successful. The agent-based framework lends itself well to the complex nature of forest fires. Integrating complexity at the level of a disaggregated fire line means fire behaviour emerges from the bottom up, as in physical models, but with far less computational load. While the fire-wind feedback mechanism has a role in adequately simulating fire

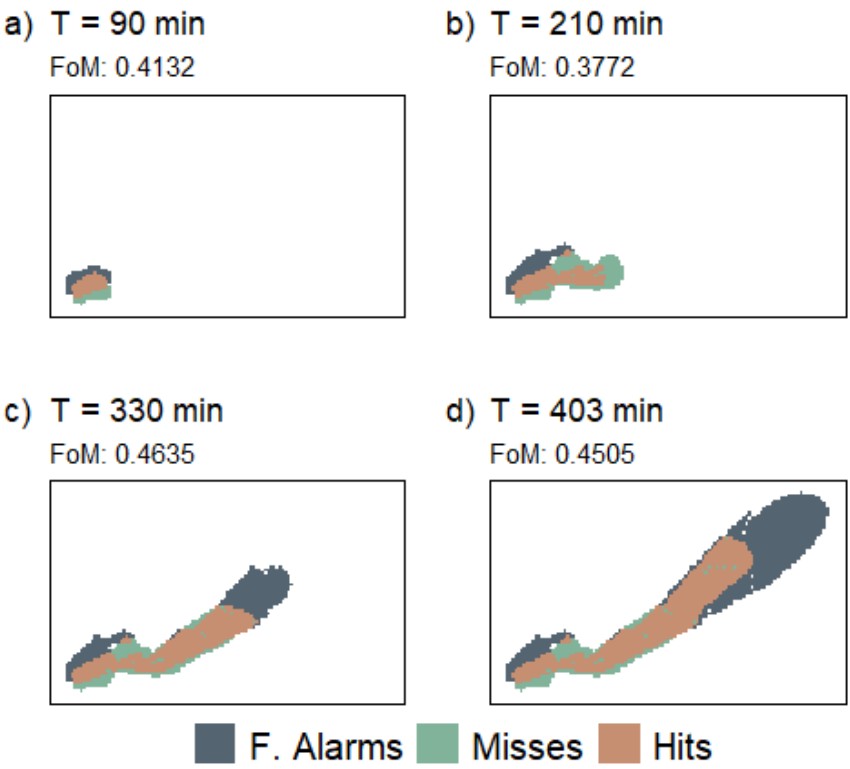

**Figure 8.** Components of FoM for case 3, at four times past ignition.

behaviour, it is clearly not the sole factor at play, both as a source of error and a vital part of successful simulation. Despite the simplicity of the fire-wind feedback sub-model, the results presented in Table 2 and Fig. 5 do indicate that ABWiSE produces more realistic simulations with it than without it. Even though the results of the validation are promising, there is still room for improvement. Two avenues for improvement are, of course, more data and an enhanced model. Exploring the limitations of both the data and the model helps us by highlighting the successes and failures of this approach, and guides future work.

## 5.1 Error and data limitations

Differentiating between input error and model error requires high quality data to minimize input error, leaving the model as the only potential source of error. Data availability and quality limits the validation of the model, in particular the weather observation data for case 4. Simulation of case 4 by Prometheus has a FoM of 0.21 (compared to the Dogrib perimeter at a 200 m resolution), because it over-burns a large area southwards, which indicates that case 4 is difficult and complex to simulate, and reinforces the notion that input data for it are inaccurate and a large source of error. However, we can consider our test cases 1 and 2 to have perfect input data, since the comparison was another model's output based on the same data. Any inaccuracy in cases 1 and 2 is due to model error. The data for case 3 is the best real-world data available to us, and is of sufficient quality

**Table 3.** Figure of Merit values from ensemble simulations for different fuel types.

| Scenario<br>Fuel type & wind speed | Figure of Merit | | Hits | | Misses | | False Alarms | |
|---|---|---|---|---|---|---|---|---|
| | Mean | SD | Mean | SD | Mean | SD | Mean | SD |
| C1 00 km | 0.16 | 0.01 | 3 | 0 | 0 | 0 | 16.37 | 0.86 |
| C1 20 km | 0.27 | 0.02 | 33.69 | 2.04 | 89.31 | 2.04 | 1.57 | 1.32 |
| C2 00 km | 0.83 | 0.02 | 56.96 | 0.2 | 0.04 | 0.2 | 11.35 | 1.28 |
| C2 20 km | 0.82 | 0.06 | 146.11 | 9.23 | 15.89 | 9.23 | 15.86 | 8.52 |
| C3 00 km | 0.11 | 0 | 9 | 0 | 0 | 0 | 72.46 | 1.91 |
| C3 20 km | 0.54 | 0.04 | 105.29 | 4.02 | 1.71 | 4.02 | 89.4 | 16.89 |
| C4 00 km | 0.79 | 0.02 | 64 | 0 | 0 | 0 | 17.45 | 1.97 |
| C4 20 km | 0.79 | 0.06 | 168.77 | 9.4 | 18.23 | 9.4 | 28.49 | 13.81 |
| C5 00 km | 0.1 | 0.01 | 2 | 0 | 0 | 0 | 17.39 | 1.01 |
| C5 20 km | 0.61 | 0.03 | 23.73 | 0.51 | 3.27 | 0.51 | 11.94 | 2.46 |
| C6 00 km | 0.46 | 0.02 | 9 | 0 | 0 | 0 | 10.43 | 0.87 |
| C6 20 km | 0.22 | 0.01 | 34.42 | 1.79 | 118.58 | 1.79 | 0.87 | 1.31 |
| D1 00 km | 1 | 0 | 1 | 0 | 0 | 0 | 0 | 0 |
| D1 20 km | 0.17 | 0.01 | 1 | 0 | 0 | 0 | 5 | 0.4 |
| M1 00 km | 0.84 | 0.04 | 19.3 | 0.83 | 3.7 | 0.83 | 0 | 0 |
| M1 20 km | 0.62 | 0.03 | 34.93 | 1.98 | 21.07 | 1.98 | 0.77 | 1.05 |
| O1 00 km | 0.8 | 0.03 | 21 | 0 | 0 | 0 | 5.18 | 0.95 |
| O1 20 km | 0.84 | 0.05 | 36.32 | 1.17 | 1.68 | 1.17 | 5.17 | 2.84 |
| Case 3 Rand | 0.04 | 0.02 | 104.58 | 65.19 | 2318.42 | 65.19 | 0.25 | 1.35 |

for the Prometheus model to have a FoM of 0.568. On the other hand, there are obvious problems with the reconstructed fire perimeters used for the time-based validation of case 3: the reconstructed fire progression does not reach the full extent of the observed final perimeter, nor is it as wide, indicating some discrepancy between reconstruction and reality. Limits to input data do not mean model errors are the same between ABWiSE and Prometheus. Attributing sources of error and uncertainty in model output is the goal of Sensitivity and Uncertainty Analyses (SA and UA, respectively).

The preliminary sensitivity analysis pertaining to fuel types demonstrates that fuel is an important subject of error in the model. The true source of error is presently indistinguishable between the model procedures using the fuel type variables and the fuel type variables themselves. The Spread, Death, Preheating, and Consumption procedures all use or affect these variables. Furthermore, this analysis shows only the discrepancy between ABWiSE and Prometheus, not real fire behaviour. However, the fact that randomized fuel resulted in an extremely low FoM for case 3 means that fuel is an important input factor and its parameterization is at least somewhat correct. Described further in Sect. 5.3, SA and UA are the next step for the model presented in this paper.

The general problems of data limitations can be addressed by new field experiments and observation techniques (Chuvieco et al., 2019). In particular, the proliferation of publicly available satellite data is a great resource for forest fire observations, though limits to return-time and resolution affect the quality and applicability of these observations (Andela et al., 2019). Canada's future WildFireSat mission (https://www.asc-csa.gc.ca/eng/satellites/wildfiresat/default.asp, accessed June 09, 2021) will address this issue and provide daily infrared observations of wildfires at a 200–500 m resolution; an ideal scale for the niche ABWiSE aims to fill.

## 5.2 Model limitations

ABWiSE makes many assumptions about fire behaviour, in the form of the equations that define fire agent RoS and heading and their relation to environmental variables. Another assumption is the simple fire-wind feedback sub-model. There was no intention for the equations based on these assumptions to be a new way to explain fire behaviour. Rather, they were kept relatively simple in order to explore the potential of ABM as a way to simulate fire behaviour in a bottom-up, complex systems approach. The design of these equations makes use of numerous parameters so that the relations between agents and input variables could be honed in on through calibration across many scenarios. Although the equations are purely empirical in nature, not adhering to the physics of fire (thus imposing an ultimate limit on the model's accuracy and validity), the modelling approach and the calibration framework mean that the model could be continuously improved with more data up to that limit. However, the corollary to this –that the model performs well in spite of a purely empirical formulation– supports our objective of demonstrating the potential of ABM for fire spread simulation.

## 5.3 Future work

Future work on ABWiSE may focus on Sensitivity and Uncertainty Analyses. Together, SA and UA quantify the overall uncertainty of a model and partition the output variation among the input factors. These input factors include not only parameters, but data and even the model's equations and algorithms. By this process, we could clearly identify the limits of the model and attribute the uncertainty to specific sources. From this point, a renewed calibration effort could proceed on the sources (input factors) most influential to the model output. However, this would require more input data to analyse the model over a larger spread of scenarios, as well as potentially billions of simulations to properly explore the parameter space. As demonstrated with the brief sensitivity analysis presented above, examining one factor at a time is not necessarily enough to identify precise sources of error. However, if we performed similar analyses pertaining to wind and terrain, we might discern which of the major environmental inputs upon which to focus our efforts first.

Given that ABWiSE is currently a proof-of-concept model, and we consider that it has proven the concept of using ABM to simulate fire spread, a simpler way forward may be to replace many of its algorithms and equations with adaptations of empirical models. Specifically, implementing the FWI and FBP system equations in a way that accommodates the ABM approach and fire-wind interactions. This wind feedback, in turn, may be generated by coupling with a CFD, or most likely implementing the pyrogenic potential model of (Hilton et al., 2018). ABWiSE, in its current state, would then serve as the benchmark for improvements.

One of the benefits of the ABM approach, and the NetLogo environment in general, is that it is relatively easy to add functionalities to the model, such as fire suppression. For example, fire-fighting efforts are an important factor in the behaviour of fires subjected to it, and suppressed fires tend to be significant for their proximity to the wildland-urban interface (Johnston and Flannigan, 2018). In its current form, ABWiSE could simulate the effect of fire-fighting by simply reducing the flammability and/or available fuel in those cells being suppressed. The matter of simulating intelligent fire-fighter behaviour is a completely different challenge, however.

## 5.4 Computation

All simulations in this study used a desktop PC with a 12-core, 64-bit processor. On average, simulation speed is 10 time steps per second, though speed goes down as the number of agents grows very large (> 500, occasionally surpassed in case 3). The most intense scenario, case 3, runs in under 80 s, on average, which compares favourably to Prometheus' 93 s on the same computer. ABWiSE's simulation time goes down for ensemble simulations, as NetLogo can take advantage of multi-threading for simultaneous runs. The Monte Carlo simulations of all four cases at two different resolutions (800 runs, producing the ensemble maps) took approximately 40 minutes. Simulation speed varied greatly during calibration, with some parameter sets resulting in very slow speeds, and so calibration took the longest time, with each parameter sweep taking about 30 h to complete.

## 6 Conclusions

Through a complex systems approach focusing on key interactions and conceiving of fire as a set of mobile agents, this study demonstrates the potential of Agent-Based Modelling for use in simulating forest fire behaviour. We present ABWiSE, an empirically calibrated ABM of fire behaviour, which succeeds at the key goal of replicating fire shape through emergence from basic rules. We evaluate the model with a suite of perimeter comparison techniques, including a time-based method, which identify specific strengths and weaknesses in simulation results. ABWiSE is still in the early stages of development, and requires more data for both calibration and validation, which will help refine its output and determine its range of applicability. It is no replacement for existing models of fire behaviour, but rather a step in exploring a new avenue of modelling. While other ABMs of fire spread such as the Rabbit Rules model or QUIC-Fire demonstrate the potential of ABM at small scales, ABWiSE applies another formulation to large forest fires, highlighting how ABM can tract the core elements of complexity of fire across scales. By using the interactions of individual agents to simulate fire behaviour, complex patterns and behaviours emerge without specifically coding them in. We believe the use of ABM in fire modelling merits further research as it leverages efficient bottom-up simulation of complex systems for coupled fire-wind interactions.

*Code and data availability.* The ABWiSE code, along with the data used for the simulations presented in this paper, is freely available on GitHub (https://doi.org/10.5281/zenodo.4976112, Katan, 2021).

**Table A1.** Fuel type and variable values

| | Fuel type | | Model value | |
| --- | --- | --- | --- | --- |
| Code | Name | | Fuel | Flammability |
| C-1 | Spruce-Lichen | | 0.5 | 0.5 |
| C-2 | Boreal Spruce | | 0.5 | 0.85 |
| C-3 | Mature Jack or Lodgepole Pine | | 0.5 | 0.9 |
| C-4 | Immature Jack or Lodgepole Pine | | 0.5 | 0.85 |
| C-7 | Ponderosa Pine or Douglas Fir | | 0.5 | 0.2 |
| D-1/D-2 | Aspen | | 0.5 | 0.1 |
| O-1a/O-1b | Grass | | 0.4 | 0.6 |
| M-1/M-2 | Boreal mixed-wood | | 0.5 | 0.6 |
| - | Non-fuel | | 0 | 0 |
| - | Water | | 0 | 0 |

## Appendix A

### A1   Fuel type characteristics

Table A1 presents a detailed description of the variables mapped and used in our model. The Dogrib fire case study does not include all 16 fuel types of the FBP system, thus we only present those present. The mapping of fuel types to fuel and flammability values for our model uses the curves presented in (Forestry Canada Fire Danger Group, 1992). The flammability value is based on the steepness and maximum value of the Rate of Spread vs Initial Spread Index curves in section 7.2 of the aforementioned report. Fuel values are based on assumptions of fuel type characteristics. This is a gross simplification of fuel type characteristics, but the use of a simple index for each value means a sub-model could later serve to generate more accurate values.

### A2   Procedures

#### A2.1   Fire-wind interactions

The local wind vector $L$ is the weighted average of the global (or ambient) wind $G$, the effect of fire on wind $L$, and the current local wind $L_0$, written as:

$$L = w_1 G + (1 - w_1)(w_2 L_0 + (1 - w_2) F) \tag{A1}$$

where $L_0$ is the local wind based on values of the previous time step, and *w1* and *w2* are weighting parameters. Only cells within a certain distance of fire agents (6 cells if the resolution is 200 m) calculate a local wind vector, and only a subset of

these (cells within 4 cells of fire), calculate the effect of fire on wind and apply a smoothing function to their wind vectors. The smoothed local wind vector for the subset is the Inverse Distance Weighted (IDW) interpolation (Eq. A2) of $\boldsymbol{L}$ of the larger set. The general formula for IDW is

$$IDW(x) = \frac{\sum_{i=1}^{n} \frac{x_i}{d_i^p}}{\sum_{i=1}^{n} \frac{1}{d_i^p}} \tag{A2}$$

where $d$ is the distance between $x$ and $x_i$, and $p$ is a constant value affecting the influence of distance. These calculations mean that at the exterior edge of this active wind zone, global wind is the most influential factor on local wind, and fire has the strongest effect in cells with fire agents present.

The fire influence, $\boldsymbol{F}$ in Eq. (A1), is the sum of a local gradient of fire RoS, $\nabla_{RoS}$), and a smoothed fire vector, $IDW(RoS)$. The gradient $\nabla_{RoS}$) is a vector pointing to the greatest change in the sum of the RoS of fire agents in the eight neighboring cells (aka the Moore neighborhood), with the exception that if there are no fires in one of the neighboring cells, the value for that cell is substituted with that of the center cell. The value of $\boldsymbol{F}$ is then,

$$\boldsymbol{F} = k\nabla_{RoS} + IDW(RoS) \tag{A3}$$

with the constant, $k$, scaling the effect of $\nabla_{RoS}$. Because fire agents spawn and die suddenly at each time step, we used $IDW(RoS)$ of fires in that Moore neighborhood to improve continuity between time steps. This is a very simple proxy for actual fire-wind interactions and it was inspired by the pyrogenic potential of Hilton et al. (2018).

### A2.2  Fire spread

Fire agent RoS is the result of flammability, wind, and slope at its present location. Many corrective factors were necessary to match the relationship between RoS and wind speed and direction to observations, as well as producing a reasonable fire shape. In short, low wind speeds have a small effect on the fire agents, but have a stronger effect on fire agents whose heading is close to the wind direction. The relationship between RoS and wind speed follows a logistic curve based on the same assumption as (Forestry Canada Fire Danger Group, 1992) that there exists a maximum RoS based on fuel type. Equation (4) shows how fire agent RoS, wind, and slope vectors are combined to determine the new RoS by which a fire agent will move this time step, and carry on to the next.

$$RoS = RoS_b f_{mod}\rho_{mod} + \boldsymbol{L}(1.05 - f_{mod})w_{mod} + \boldsymbol{S}s_{mod} \tag{A4}$$

where

$$RoS_b = \left[ f_1 RoS_0 + (1 - f_1)(flam^{1.3} + flam + w_3\|\boldsymbol{L}\|\|collinear\|) + s_1\|\boldsymbol{S}\|coslope \right] m_1 w_{mod} \tag{A5}$$

where $flam$ is the flammability of a cell; $f_1$, $w_3$, $s_1$, and $m_1$ are user-defined parameters; $collinear$ and $coslope$ are the cosines of the difference between the fire agent's heading and the wind direction and terrain aspect, respectively. Using the absolute value of $collinear$ means that fires moving directly into the wind still increase their RoS instead of slowing to a stop.

This reflects an assumption that the oxygen supplied by the wind in this case is sufficient to increase the strength of the fire, allowing fire to move against the wind in low-wind conditions. The term $w_{mod}$ represents the logistic equation with parameters $a$, $b$, and $k$:

$$w_{mod} = \frac{a + \frac{flam}{30}}{\left[1 + be^{-k\|\boldsymbol{L}\|}\right]^{\frac{1}{flam+2.6+RoS}}} \tag{A6}$$

In Eq. (A4), $f_{mod}$ is an additional correction component with some constants fixed at values that appeared to provide acceptable model behaviour, and one parameter, $f_2$, is left open to more thorough parametrization.

$$f_{mod} = f_2(1.2 - \frac{w_{mod}}{1.3}) \tag{A7}$$

Finally, $\rho_{mod}$ is another correction factor based on the density of fire agents, representing an assumption that closely clumped fire agents stay hotter, longer, and fire agents out on their own lose heat more quickly and don't move as fast. Density, $\rho$, is 545 expressed as the number of fire agents within a radius of 1 of the agent calculating it, and the near-density, $\rho_{near}$, is the mean density of those same agents in a radius of 1, such that the density modifier is:

$$\rho_{mod} = \frac{\rho + 1}{\rho_{max}} - \frac{\rho_{near} + 1}{\rho} \tag{A8}$$

where $\rho_{max}$ is the maximum density of all fire agents at that time step.

### A2.3  Preheating

Agents heat the cell ahead of them at a distance of their RoS by raising its flammability. RoS may be less than 1, thus the "cell ahead" may be the cell the agent is already in. The modelling software determines cell location by the center of the cell, so a cell that is one RoS away may have a different distance from the agent. For example, if the edge of a cell is one RoS away, its distance to the agent is one RoS + 0.5. Therefore, the distance between the cell and the agent, $d$ in Eq. (A9), is not the same as the RoS. Only cells with a flammability below 1 (the maximum) are heated by the amount defined by:

$$flam = flam_0 \frac{0.005 RoS}{(1+d)^2} \tag{A9}$$

### A2.4  Death

Just after moving, fire agents have a chance to die out if the fuel value at their location is below a certain threshold modulated by their own RoS (Eq. A10). This means that slower fires have a higher chance to die out at higher fuel values than faster fires. If the fuel of their current cells is lower than that threshold, agents die if they generate a random floating-point number 560 between 0 and $RoS^-1$ that is less than 1. This means that slower fires, while triggering this condition sooner than fast fires, have a smaller chance of actually dying out. This counterbalancing aims to simulate a kind of smouldering behaviour.

$$fuelthreshold = 0.2(1.1 - RoS) \tag{A10}$$

$$P_{die} = ran\frac{1}{RoS} \tag{A11}$$

## A2.5   Propagation

If, after moving, fires find themselves beyond $\sqrt{4 \times RoS}$ cell lengths from their start location, and if there are fewer than 3 other fire agents already in that cell, they spawn 3 new fire agents then die. The limit of 3 prevents an excessive number of agents from suddenly appearing in one cell and very rapidly consuming all the fuel. Slower fire agents spawn and die more frequently than faster fires. The new fire agents each inherit their "parent's" RoS and heading and deviate from that heading by -45, 0, and +45 degrees, respectively. These new fire agents consume fuel on this tick, but only start moving on the next tick.

## A2.6   Consumption

Finally, the fires present at this time step of the simulation consume fuel. They reduce the fuel value of the cell they are in by $RoS \times fuel \times B^{-1}$ where $B$ is another parameter. Including the fuel variable in the rate of consumption means that cells with high fuel levels lose fuel quickly, but as fuel reduces, it burns away more slowly.

*Author contributions.*   Author 1 conceived, developed and evaluated the ABWiSE model, carried out the simulations and analyses of the model, produced the figures, and wrote the manuscript with the support of Author 2. Author 2 helped design the study, supervised the project and helped write the paper. Both authors discussed the results and contributed to the final paper.

*Competing interests.*   The authors declare that they have no conflict of interest.

*Acknowledgements.*   The research has been supported by the Natural Sciences and Engineering Research Council (NSERC) of Canada Discovery Grants awarded to the second author (grant no. RGPIN/05396-201) The authors would also like to thank Saeed Harati and Andy Hennebelle for their invaluable feedback throughout the writing of this manuscript.

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
