# Peer review of "ABWiSE v1.0: Toward an Agent-Based Approach to Simulating Wildfire Spread"

_Natural Hazards and Earth System Sciences, 2021_

## Community Comment (CC1)

Figure 3 alternative color schemes:

[Figure]

Option 1) colours based on FBP system colour scheme

[Figure]

Option 2) Technically coherent colour palette.

---

## Author Comment (AC1)

Supplement to Author reply of RC2

Revised Figure 1.

---

## Author Comment (AC2)

Option 1) colours based on FBP system colour scheme

[Figure]

Option 2) Technically coherent colour palette.

---

## Author Response (AR1)

We are very grateful to the referees for their invaluable feedback. We strive to address the referees' comments below. In our response, the original comments are in italics, and our responses follow. Changes to the manuscript are indicated by line number of the tracked changes version, and are listed in bold font in this response.

**Response to reviewer 1**

1. *The introduction needs acknowledgement of anthropogenic forest fires. This needs to cover the fact that controlled burning has been used for a long time (and effectively) as a forest management tool and the intensification of forest fires due to climate change which is also anthropogenic.*

The authors would like to thank the reviewer for this comment and agree that this information needs to be included in the Introduction. **The new information can be found starting on line 15 of the tracked changes version of the revised manuscript.**

2. *Good summary of fire models. However, the jump to the models is a bit abrupt. A mention of satellite-based forest fire monitoring approaches would help. Here is a great example using MODIS: https://modis-fire.umd.edu/*

The authors welcome this excellent suggestion and propose to add a paragraph to integrate this information. In addition to providing a smoother transition, the implementation pf this suggestion supports our brief discussion of the value of remotely sensed fire observation data for model validation. **The segue to fire models by way of satellite-based forest fire monitoring has been added to the start of Section 1.1, as of line 45**.

3. *"Ultimately, the goal of such a fire simulation model is to predict fire behavior, but presently, the purpose 150 of ABWiSE is to explore how ABM, using simple interactions between agents and a simple atmospheric feedback model, can simulate emerging fire spread patterns." – This needs further explanation. How is predicting fire behavior different from simulating emerging fire spread? What is ABWiSe achieving and what is it leaving out. What are the merits and demerits of the trade-off?*

The purpose of the statement quoted above was to indicate that ABWiSE is currently a proof of concept, and should not be used for fire behaviour prediction where the results may impact decision-making. **In the revised manuscript, we have adjusted the phrasing of the statement starting on line 162, to better specify what the goal of the study, and ABWiSE, is**.

Perhaps a point of confusion is the use of the term "emerging", which we use in reference to the emergent properties of a complex system, rather than "a prediction of what unfolds". Ultimately, there is no trade-off between the goal of simulation and the goal of prediction, because if we can simulate well, we can predict well. The caveat, and the reason we focus on demonstrating the value of ABM for simulating forest fires, is that by eschewing physical simulation in favour of empirical calibration, we gain rapid computation but cannot confidently extrapolate our model to new scenarios without further validation. So, yes, we can predict fire behaviour through simulation for the fires ABWiSE is calibrated to, but cannot yet ascertain the validity of predictions for other fires.

4.  *Line 185: FWI seems to provide a great resource to test model assumption regarding wind speed. If it is not too complex to incorporate within the current study, this will be a good addition to the paper. Also, is the relationship between RoS and wind speed mentioned in line 205 the same as used in FWI?*

The FWI provides an index of fire danger based on fuel moisture, temperature, and wind, but not fuel type. Rate of spread estimates are provided by the Fire Behaviour Prediction (FBP) system that uses the FWI equations for some inputs. This system has been in use for decades, but does not account for any form of feedback mechanism. The way ABWiSE uses feedback loops to replicate fire behaviour means it is difficult to isolate one particular assumption at a time. In particular, ABWiSE's variables for fuel availability and flammability are similar to the Buildup Index and the Fine Fuel Moisture Code components of the FWI, respectively; however, for the latter pair, the FFMC takes wind speed as an input, as it forms a component of the Initial Spread Index (another component of the FWI), while flammability in ABWiSE does not. Thus, using the FWI to supply fuel availability and flammability values for ABWiSE would have required a reworking of the FWI without wind.

While it would be a great idea to take the FBP system apart and recombine it for use in a complex ABM where feedback plays a major role, we deemed it beyond the scope of this study. **We have addressed this issue on lines 200 to 204.**

As for testing model assumptions regarding wind speed against the FWI, in essence, wind speed assumptions are tested in whole with the perimeter comparison methods. Prometheus uses the FBP and FWI to calculate the RoS for its fire perimeter, and we calibrated ABWiSE against two base cases generated by Prometheus. We considered performing perimeter comparisons between Prometheus' simulations of the Dogrib fire, but chose not to, since the goal is to simulate a real fire as well as possible, not simulate a simulation as well as possible.

The relationship between RoS and wind speed is not the same as used in the FWI, though it makes use of the same form of equation. However, the various feedbacks change the final RoS from that particular equation, so it would be moot to use the exact same relationship between wind speed and RoS as the FWI. **We have clarified the difference between wind speed and RoS relationships starting on line 227.**

5.  *How does ABWiSE improve upon Prometheus?*

In short, by incorporating feedback mechanisms while using similar inputs appropriate for a Canadian context. However, we make it clear that ABWiSE is no replacement for Prometheus, but rather a step in a new direction that may lead to models that could improve upon Prometheus in many ways. We believe the discussion provides sufficient information about the value of incorporating feedback mechanisms for fire behaviour simulation. **However, changes to Table 2 and to line 474 in Section 5.4 Computation, provide some direct comparisons between ABWiSE and Prometheus.**

6.  *Can the model be used to simulate fire fighting efforts? For example, given that forest fires can span thousands of acres, what happens when fore fighting efforts are concentrated in some area as opposed to another?*

Simulating firefighting efforts is absolutely possible by coupling ABWiSE with a model that could determine *where* firefighting efforts would occur. One possible way to incorporate firefighting would be to simply reduce the flammability and fuel availability values in those cells being suppressed. **This is an interesting topic and we have added it to our discussion of future work, starting on line 465.**

*7.   Figure 3: Great map. Would it be possible to use a different color scheme to better delineate the different vegetation types?*

**We would like to thank the reviewer, and we have altered Figure 3 with a new colour scheme.**

*8.   The 4 cases can be presented in a table to improve readability. I kept having to find them in the text to remind me exactly what they were.*

**The authors would like to thank the reviewer for his suggestions and have done so in the form of the new Table 1.**

Lastly, the referee suggests we make our code and data available for use and improvement by the broader research community. We have made it available as a GitHub repository, and included the information to access it in the *Code and data availability* section of the manuscript. **We have also placed this information earlier in the manuscript, on line 192.**

**Response to reviewer 2**

**What is 'Agent-Based'?**

*As an aside before my more substantive comments, a personal concern I'd like to raise here is the presentation of model as 'agent-based'. The term is very fashionable and because of this is often applied when a better term would be appropriate. For example, as Bithell et al. (2008, doi: doi:10.1016/j.geoforum.2006.10.014) distinguish discrete-element, individual-based and agent-based modelling. These are all essentially the same approach - aiming to "represent the interactions of individuals or entities with one another and their environments by sets of computational rules" with roots in complexity science. This is also the case for ABWiSE. The differences between the terms is disciplinary and dependent on what type of individuals are being represented; 'discrete-element models' are used in geomorphology, 'individual-based models' are used in ecology and 'agent-based models' are used in social science (where the individuals are understood to have some kind of agency or decision-making capacity). To my mind ABWiSE would be better described as a discrete-element model, where discrete elements are akin to 'flames' (rather than 'fires' as used in the model code) as I tend to understand agents as having some kind of decision-making agency. But I do recognise that 'agent' has become de rigueur and my personal view is a backburn that will be overwhelmed by the conflagration, so my comment here is more for public reflection than any change in the manuscript.*

The authors are thankful to the referee for bringing up this point for public reflection. We believe there is value in using the term "agent-based modelling" in a broader sense, not only because it is becoming *de-rigeur*, as the referee has said, but also because it invites originality. Used as a catch-all term, it can make it easier to compare similar methods across disciplines, and can encourage researchers to consider new problems under the lens of complex systems science and agent-based modelling. As to agency being a prerequisite of ABM, the Boids model is certainly considered an ABM, though the agency of the agents is very limited. If perception and (re)action are the chief elements of agency, then the term can encompass a lot. While ABWiSE could indeed fall under discrete-element methods, the concept of the model stemmed more from this concept of agency than discretization, which is why we prefer to use that terminology.

**Complex vs Complicated**

*However, I do think there are some points in the manuscript where the language and concepts use to frame the development of ABWiSE in complexity science need changes. For example, I think we need to be careful about distinguishing complexity of model behaviour from the complicatedness of the model structure. This is well explained and explored by Sun et al. (2016; doi: 10.1016/j.envsoft.2016.09.006). There are some points in the manuscript where this distinction is blurred and I think the authors need to clarify, particularly given the emphasis they place on how their model approach aims to exemplify concepts from complexity theory and given this is a proof-of-concept paper (so we need to be clear about the concepts). For example:*

- *Line 51 I think you actually mean that the gap is between complicatedness of model structure and the speed of model execution? A very simple model could run very quickly and but still produce complex behaviour (i.e. complexity, e.g. the classic boids flocking model).*
- *Line 58, isn't the problem with CFD models that they are very complicated (needing many calculations) rather than they inherently produce complexity?*
- *Line 63 'small' I think you mean 'individual' here? It's not the size of the entity that is important for modelling complexity, rather the that there interactions of individual (discrete) entities that might give rise to emergent phenomena*
- *Line 129 I don't think we should be aiming to build 'complex' models. Rather, we should aim to develop simple (enough) models that enable us to understand complex phenomena (although often our models do become quite complicated). Can you check your use of language (complex vs complicatedness) and if you really do want to build a model with a complex structure, justify why that is better than one with a simple structure that produces complex behaviour (which I think is actually what you are trying to do).*
- *Line 6: given my points above, please consider whether you really do think physical models are necessarily complex and if this is really what you are hoping to combine with empiricism*

The authors are grateful to the referee for bringing these points to our attention. Overall, we agree that a better distinction between complicatedness and complexity will improve the quality of the manuscript. We have frequently used the term complex model to mean

a model that represents complexity, rather than a model with complex structure. We have revised these instances to clarify that distinction.

On point one, while it is true that there are many simple (and fast running) models that produce complex behaviour, the preceding discussion in the manuscript notes that physical models typically represent the most complex behaviours, but they run slowly because of the way they do so. The gap between complexity and speed we were referring to is specific to fire simulation models, and more precisely those we have included in our Background section. **We have changed the phrase on line 64 to clarify this.**

On point two, indeed, that is poorly phrased. **We have removed the word "complex" on line 71 and replaced it with "complicated".**

On point three, we meant as a small part of the whole system, rather than physically small, but we agree that individual may be a more suitable term. **We have changed "small" to "individual" on line 76.**

For point four, we do not wish to be creating complex models, as such. The intention was to mean models that account for / include the relevant complexities of the system at hand. We have revised the manuscript to ensure there are no references to "complex model structure" and that those phrases indicate "models that have complexity", instead. **The explicit changes not addressed in response to other points can be found on lines 35 and 142.**

The last one is an important point. We intended to show that, in what we have reviewed, physical models have typically been the only ones able to successfully address the complexity of forest fires, especially fire-wind interactions. Most of the physical models *are* complicated, however, and it is this complicatedness we wish to reduce using empiricism and ABM. **The phrasing is addressed on line 6, and other changes in response to this comment as a whole clarify the focus on complexity of model behaviour rather than complexity of model structure.**

***Model Flow***

*In general, the model is well very described. The appendices seem to contain all procedures and they are clear. The main exception to this for me is Figure 1 which is a flow chart to 'describe model procedures'. This is very difficult to read and needs to be improved. First, what do the arrows represent? Do they show the flow of time (order of execution) or indicate flows of information? Or both? Or something else. Maybe different types of arrows are needed (then a key is also needed). A key (legend) to explain the different shapes of boxes would be useful. Really think you should aim here to represent the order of execution of the model and, to do this most explicitly, every equation listed in Appendix A2 would be shown in the flow chart so that readers can understand in what order they are executed (without needing to check the source code). I understand that some procedures here are 'global' (once per iteration) while some are for multiple agents (so repeated multiple times per iteration), but some use of symbology can help here.*

We thank the referee for pointing out the shortfalls of Figure 1. **We have provided a revised version of Figure 1.** It is more explicit about the order of execution, where procedures are arranged chronologically left-to-right, and the "inner loop" performed by every agent within a model time step is clearly identified. The box shapes follow standard convention for data (the cylinders), procedures (rectangles), and choices (diamond), so we thought the distinction was clear. The arrows show chronology.

**Fuel Types**

*The approach you take to calibrate your model is sensible and seems to work relatively well. As you highlight it's not feasible to examine all possible parameter combinations and the CART approach seems to works well. However, it seems to me that even though the fuel variables are inputs to the model, they are still a key part of 'the model' (in the sense that they are conceptual representations of what will burn) and should be subject to some assessment. As you note in the appendix the fuel values are gross simplifications - this suggests some kind of sensitivity analysis of their values would be worthwhile. It would require more model runs, but not necessarily too many. Furthermore, I wonder if analysing results (for scenarios 3 and 4) relative to fuel type would be useful to understand errors. For example, a simple set of box plots for hits, misses and false alarms by fuel type might highlight whether some fuel types are poorly parameterised. However it is done, and while I appreciate you intend to do more comprehensive sensitivity and uncertainty analyses in future, I think some consideration of the uncertainty in the modelling due to fuel variable values is needed as this is a key input to the model.*

The authors thank the reviewer for this suggestion. While we agree that some analysis of the fuel type values is important, the method of analysis suggested by the reviewer may not be suitable. Given that fire propagates along the landscape in the simulation, and that different fuel types are distributed throughout, errors resulting from any poorly parametrized fuel type could result in error in other fuel types. We propose, instead, to test the different fuel types using scenarios 1 and 2. This would allow us to compare ABWiSE's response to fuel types with that of Prometheus. **We have performed experiments to evaluate the model's response to fuel types and present the results in Table 3. Corresponding additions to the text are found on lines 268, 382, 425, and 455.**

**Wind**

*The incorporation of 'fire-atmosphere' feedback is good. But there does not seem to be an explicit consideration of how important incorporating this feedback is. Some consideration of this would be useful and it could be as simple as running the model with and without the feedback included to show the difference in performance (e.g. for the four scenarios). Furthermore, as it currently stands in this manuscript I think 'fire-atmosphere feedback' is a bit of a misnomer and could be explained more simply if it were described as fire-wind feedback (akin to how you name it in the source code). I can see that in future you may include other aspects of atmosphere but here you are solely looking at a 'fire-wind feedback' so why not call it that? (especially in Figure 1).*

The referee makes an excellent point that the value of incorporating fire-wind feedback is never explicitly considered in our manuscript. **We have performed tests without the wind feedback, as suggested, as well as changed our terminology from fire-atmosphere feedback to fire-wind feedback, as appropriate. Elements pertaining to the wind feedback tests are found on lines 337, 343, and 405, and in Table 2 and the new Figure 5.**

**Model Comparison**

*I would like to see more explicit comparison with the Prometheus model (and maybe others). You do mention FoM values for Prometheus in the text but including these values in Table 1 would enable to reader to compare performance of ABWiSE v1 directly. I don't think you should be shy in doing this. I agree that in it's current state ABWiSE 'is no replacement for existing model' but why couldn't it be in the future? The only way to advance the model with that objective is to directly compare model performance. Furthermore, I was encouraged to see that the model executes quickly (section 5.4) and so could feasibly be used operationally, but how does speed of execution compare to Prometheus (i.e. compare practicality as well as model performance). As a side note, if you were to refactor the model in a language faster than NetLogo (e.g. maybe using AgentPy in Python, https://agentpy.readthedocs.io) then this would be even more feasible. Ultimately, my comment here is that even though this is a 'proof of [the agent-based] concept' manuscript, I would like to see more direct comparison of ABWiSE with existing models.*

We thank the referee for their suggestions on model comparison. **We have included FoM comparison data for Prometheus in Table 2, and comparisons of execution time in Section 5.4**

Refactoring in another, faster language is an ultimate goal for ABWiSE, though for now, NetLogo is ideal for quickly implementing changes (due to its simple syntax and the modellers' familiarity with it). It is also worth noting that the current code in NetLogo has not been optimized for speed, so there may be room for improvement there, too.

**Specific comments:**

- *Table 1: Why do you present Standard Deviations and not Standard Errors? I think the latter would be more useful for comparing performance between the various cases.*

Table 2 presents the distribution of FoM for the ensemble simulations, and as such is descriptive, rather than predictive. These distributions have no expected predictions from which to have errors, and so we report the standard deviation to describe the spread of possibilities in the ensemble of simulations. **We have specified that the table shows descriptive statistics in the caption for Table 2.**

- *Line 315: I am unclear why case 3 is highlighted as having particularly low performance at the coarse resolution? In Table 1 it looks to me that Case 4 has worst absolute performance and case 3 actually has the smallest absolute decrease in Mean FoM between 200m and 500m resolutions. So this sentence confuses me. Could you please clarify*

We thank the referee for noticing this discrepancy. Upon reflection, the statement is not particularly relevant to the manuscript. **We have removed the statement on line 342.**

- *Figure 4: I am confused about what 'Grid Spacing' is, as referred to in the figure caption. Grid spacing is not mentioned elsewhere in the manuscript. Could you please clarify.*

We are referring to the gridlines (graticules) used to represent scale in each of the sub-figures. We thought it important to mention because there was no room to include the coordinates, as in i), for the rest. We thought it was important for the reader to be able to compare sizes between cases. **We have changed the terminology of Figure 4's caption, and added a statement explaining the purpose of the graticules.**

- *It is great that your model is open source with freely available source code. If the journal rules allow I suggest you make this clearer earlier in the manuscript (not leave right at the end). For example, I suggest you make a reference to the code on Line 178 where you state the implementation of the model.*

We thank the referee for this suggestion, especially for including where to put the reference. **We have added the information to line 192, as suggested.**

We hope that our responses and the changes to the manuscript satisfy all the reviewers' comments and suggestions.